# The Effect of OMMT on the Properties of Vehicle Damping Carbon Black-Natural Rubber Composites

**DOI:** 10.3390/polym12091983

**Published:** 2020-08-31

**Authors:** Wei Liu, Lutao Lv, Zonglin Yang, Yuqing Zheng, Hui Wang

**Affiliations:** 1School of Polymer Science and Engineering, Qingdao University of Science and Technology, Qingdao 266042, China; qdlorin@qust.edu.cn (W.L.); tao19961225@sina.com (L.L.); gaofenzi@qust.edu.cn (Z.Y.); min@qust.edu.cn (Y.Z.); 2Key Laboratory of Rubber-Plastics, Ministry of Education/Shandong Provincial Key Laboratory of Rubber and Plastics, Qingdao University of Science and Technology, Qingdao 266042, China; 3Department of Chemical Engineering, University of Waterloo, 200 University Avenue West, Waterloo, ON N2L 3G1, Canada

**Keywords:** organic montmorillonite, natural rubber, damping properties, mechanical properties

## Abstract

In this study, the filled natural rubber (NR) was prepared with organic montmorillonite (OMMT) and carbon black (CB). The effects of the amount of OMMT on the properties of CB/NR composites were investigated by measuring the physical and mechanical properties, compression set and compression heat properties, processing properties and damping properties. The formulation was optimized depending on the different conditions of end applications and the damping properties of rubber were maximized without affecting the other properties of the rubber. The results showed that the rubber composite system filled with 2 phr (parts per hundreds of rubber) OMMT had better mechanical properties and excellent damping performance.

## 1. Introduction

With the development of rail transit, it has become a major problem to prepare high-performance (high strength, high elasticity, high flexibility and high damping) rubber shock absorbers to improve NVH (noise, vibration and harshness) performance. Rubber can transform the mechanical energy to heat energy, and so can reduce or even eliminate the vibration, and has been widely used as a damping material [1,2]. In recent years, the improvement of damping properties of natural rubber (NR) has become a hot topic [3]. It was found that damping vibration system can be obtained by changing the rubber shock absorber damping performance to improve the performance of rubber, which provides a theoretical support for later research and development of rubber damping products. Gao et al. [4] blended butadiene rubber (BR) with NR to design the new formulation of automotive shock absorber rubber products, and obtained better damping performance of composite rubber materials. Some researchers [5,6] used a grafting method to improve the rubber macromolecule chain branching degree, by which the molecular chain of internal friction was raised and then the performance of damping materials was increased.

Wang [7] reported that when increasing the content of the short fiber filling, the peak value of the loss factor Tanδ of the NR composites increased from 0.2 to 0.44. The effects of sulfur and accelerator, rubber alloy and carbon black (CB) on damping rubber were also studied extensively [8,9,10,11,12]. The amount of carbon black and the dispersion of carbon black in the material can impose significant impact on the damping properties of the material. It was also found that graphite can make a great effect on the damping performance [13]. When 5 phr graphite was added to nitrile rubber to replace the same weight of carbon black, the storage modulus of nitrile rubber increased correspondingly, and the effective damping temperature range expanded, which are beneficial to improve the damping performance. Other studies showed that CNFs provide much stronger reinforcement than carbon black [14]. Noor Azammi [15] filled kenaf fiber into TPU-NR composites and got a sample that would shift the damping temperature range up to 135 °C. Therefore, the wider the applicable temperature range, the better the damping property of the material. Joseph et al. [16] studied the changes of the damping parameters when adding palm oil microfibers and long fibers to acrylonitrile butadiene rubber (NBR). The damping parameters includes the loss factor peak, glass transition temperature (*T*_g_), storage modulus, and the damping properties of composite materials. The results showed that the storage modulus increased with the increase of the dosage and the loss factor decreases gradually. Guo et al. [17] reported that when filled with micro glass flake, the storage modulus of NBR increased in low-temperature stage and the mechanical properties and high-temperature performances almost kept well. Perera et al. [18] grafted methyl methacrylate (MMA) on the molecular chain of NR, which could greatly improve the damping performance of rubber. Xu et al. [19] prepared a rubber composite with excellent properties of improved mechanical property, high damping value and wide damping temperature range by using multilayered structure material and revealing the damping mechanism at the molecular level and in a quantitative manner.

Organic montmorillonite (OMMT) has attracted extensive attention due to its good reinforcing effect, low price, abundant reserves, simple preparation and environmental protection. As a result, it is used as a new reinforcing filler in rubber. The influence of OMMT content on swelling behavior was investigated, showing that it can significantly improve the delayed expansion property of rubber [20]. Chen et al. [21,22] found that the physical properties of rubber composites could be enhanced by OMMT, which shed light on the application of structural damping materials in the future.

It is rarely reported that OMMT was used to improve the vibration damping performance of rubber, while the layered structure of it is worth studying. At the same time, it can be found from the previous research that the damping performance of rubber is mainly affected by the structure of rubber and its filler. In this study, the processing performance, mechanical properties and damping properties of rubber composites were all improved only with little amount of OMMT. To characterize the damping performance of rubber material with OMMT (widely used as nanofillers in polymeric composites), the OMMT/CB/NR composites were obtained by mechanical mixing reaction intercalation method. In addition, studies on the influence of OMMT content and the composition of rubber components on its mechanical, dynamic mechanical properties and damping properties were performed. The characterization of the samples was done by rubber processing analysis (RPA), dynamic mechanical analysis (DMA), scanning electron microscope (SEM) and the mechanical analysis instruments.

## 2. Experiment Section

### 2.1. Materials and Preparation

Natural rubber (SCR, Hainan Natural Rubber Company’s product, Haikou, Hainan, China) was used as basis material. Sulfur (99.5% purity) was purchased from Jinchangsheng Co., Guangzhou, Guangdong, China. Organic montmorillonite (OMMT, 85%, organic modified by quaternary ammonium salt, particle size 44 μm, colloidal viscosity 4.0 mPas∙s) was purchased from Zhejiang Fenghong clay chemical Co., Anji, Zhejiang, China. Cabon Black (N330) was purchased from Cabot Corporation, Tianjin, China. *N*-cyclohexyl thio-phthalimide (CTP-70GR, a kind of scorch retarder), *N*-1,3-dimethylbutyl-*N*’-phenyl-p-phenylenediamine (4020, a kind of antioxidant), Poly (1,2-dihydro-2,2,4-trimethyl-quinoline) (RD, a kind of antioxidant) and Zinc oxide (ZnO-80) were purchased from Ningbo Actmix Rubber Chemicals Co., Ltd, Ningbo, China The remaining ingredients were provided by Jinchangsheng Co., Guangzhou, Guangdong, China. Table 1 shows the specific formula proportion of samples.

The formula of this study was listed in Table 1, on the basis of weight. In the preparation process, all raw materials were weighed according to the formula ratio, and the actual weight of NR was 200 g. Firstly, NR was dried at 60 °C for 4 h before use. Then, it was processed with stearic acid and microcrystalline wax in sk-160B two-roll mill (Shanghai Plastics and Rubber Machinery Co., Shanghai, China) at room temperature, the initial distance between the two rollers was 2 mm, completely encased on the roller for 2 min. Next, the CTP (*N*-cyclohexyl thio-phthalimide, a kind of scorch retarder), 4020 (*N*-1,3-dimethylbutyl-*N*’-phenyl-p-phenylenediamine, a kind of antioxidant), RD (poly (1,2-dihydro-2,2,4-trimethyl-quinoline), a kind of antioxidant), mixed for 3 min. The following were carbon black N330 and OMMT, mixed for 3 min. After that, sulfur and ZnO (Zinc oxide) were added into the blended compound. Next, set the distance between the two rollers to 1mm, made the “triangle bag” in the process for 5 times. Lastly, the compound obtained was molded into sheets in a Type Plate Vulcanizer (XLB-400 × 400 × 2 50 T) (Qingdao Yadong Machinery, Qingdao, China) at 150 °C for 15 min.

### 2.2. Measurements

#### 2.2.1. Processing Performance

Mooney viscosity test was carried out on a CL-2000G Mooney Viscosity Meter at 100 °C (Jiangdu Rectify Test Machine Factory, Yangzhou, China) according to the standard of ASTM-D1646. Vulcanization performance was conducted using a MDR2000 Rotor Rheometer at 150 °C (GOTECH Testing Machines Inc., Dongguan, guangzhou, China) according to the standard of ASTM-GB/T 16584-1996. Processing performance of each content were tested with three samples.

#### 2.2.2. Mechanical Tests

Tensile properties, tear strength and elasticity modulus were determined using AT-7000M Universal Electronic Tensile Machine (GOTECH Testing Machines Inc., Shanghai, China) at a tensile speed of 500 mm·min⁻^1^ according to HG/T 3849-2008, GB/T 529-2008 and HG/T 3321-2012. Hardness was measured by GT-GS-MB Shore A Hardness Tester (GOTECH Testing Machines Inc., China) according to GB/T 531.2-2009. Compression set was tested according to GB/T 7759.1-2015 (homemade equipment, type A sample, the compression ratio of sample of 25%, time for 72 h). Compression heat was tested by GT-RH-200 (GOTECH Testing Machines Inc., China) according to GB/T1687.3-2016. (Resilience was carried out using a CJ-6A Rubber Rebound Testing Machine (Shanghai Chemical Machinery No. 4 Factory, Shanghai, China) at an impact speed of 14 m·s^−1^ according to GB/T 1681-2009. Five samples of each content were tested for mechanical properties.

#### 2.2.3. Rubber Processing Analysis (RPA)

Dynamic rheological measurements of unvulcanized rubber compounds were carried out on a RPA8000 Rubber Processing Analyzer (GOTECH Testing Machines Inc., China) by a shear mode at a temperature of 60 °C, a frequency of 1 Hz, and a strain amplitude in the range of 0.28–200%, according to ASTM D 6204. Three samples of each content were tested for data of RPA.

#### 2.2.4. Dynamic Mechanical Analysis (DMA)

The DMA was carried out using a DMA 242 Dynamic Mechanical Analyzer (NETZSCH Company, Selb, Germany) in a double cantilever beam deformation mode within a temperature range of −80 to 80 °C with a heating rate of 3 °C/min and at a fixed frequency of 1 Hz. The experiment process was carried out in the atmosphere of nitrogen. Three samples of each content were tested for data of DMA.

#### 2.2.5. Scanning Electron Microscope (SEM)

The micromorphology of OMMT in rubber matrix was investigated using a JEOL JSM-7500F scanning electron microscope (SEM) (JEOL Ltd., Tokyo, Japan). Resolution: 1.0 nm at 15 kV; 1.4 nm at 1 kV. Acceleration voltage: 0.1–30 kV. One sample for each content was tested for SEM.

## 3. Results and Discussion

### 3.1. Vulcanization Performance

Normally, the change in torque during vulcanization is proportional to the density of cross-linking including the physical and chemical cross-linking. The vulcanization properties of each content were tested. It can be seen from Figure 1 that the minimum torque (M_L_) increased gradually with an increase in the content of OMMT. It was determined by the viscosity of the rubber compound, which indicates the strength of physical interaction when the rubber is not cross-linked. Therefore, when the content of OMMT was increased, the physical interactions of the rubber gradually increased. The maximum torque (M_H_) increased appreciably with addition of 2 phr OMMT. However, the increment was not obvious with more amounts of OMMT addition. M_H_ depends on the filler and the vulcanization system, and it reflected the superposition of physical interactions and chemical interactions after cross-linking. When OMMT was used in rubber, the MH-ML of compound increased, and the viscosity became larger thereby producing the agglomeration effect. Therefore, the larger content of OMMT, the stronger the agglomeration effect, and thus it damaged the mechanical properties of the material.

### 3.2. Mechanical Properties

Table 2 shows the effect of OMMT addition amount (on the basis of weight) on the mechanical properties of vulcanized CB/NR polymer composites. It can be seen that the OMMT content has an adverse effect on the physical and mechanical properties of the material, especially on the tearing strength. However, the tensile strength decreased by less than 11.5%, which still met the requirements of automobile vibration reduction products. According to the tensile strength at different elongations, it can be seen that the modulus had a decreasing trend with increasing of the OMMT contents. It is worth mentioning that the composite obtained has a greater resilience by adding 2 phr OMMT, which is similar to the variation trends of M_H_ and MH-ML as discussed in the above analysis.

Hardness was measured at five different locations of each sample. Figure 2 shows the hardness of composite material changing with the amounts of OMMT. It can be seen that the hardness values of vulcanized rubber filled different contents of OMMT were all within 65 ± 2 and the general trend of it is increase slightly with more contents, but not all of it fits this pattern due to experimental mistakes. It also reflects its effect on the packing-packing network of vulcanized rubber. These data indicate that OMMT has minor influence on the vulcanized rubber hardness, which is of great importance to the formulation design of shock absorbing products.

It can be seen from Table 3 that the final temperature rise of compression heat generation was the lowest with addition of 2 phr OMMT, and it increased gradually with increasing of OMMT amounts. When the rubber endures the cyclic stress, the macromolecular chain segment can generate relative motion, which then converts the mechanical energy into the thermal energy. It was preliminarily expected that 2 phr OMMT will reduce the agglomeration effect and improve the dispersion of CB. When excessive content of OMMT was added, the fillers were more unevenly dispersed. The filler-filler network thus became stronger and the friction between them caused the heat of rubber increased. At the same time, the static compression permanent deformation of polymer composites reduced gradually. This may be a result of the insertion of the rubber macromolecular segment into the OMMT sheet structure, which then increases the physical cross-linking and reduces the static compression permanent deformation.

### 3.3. Dynamic Mechanical Performance

In general, polymers are not used in their pure form, which means that the CB, silica and OMMT play an important role in the polymer reinforcement. The fillers in polymer strongly modifies the viscoelastic behavior, especially the dynamic mechanical properties of the composite. In dynamic mechanics measurements of shear modulus, the Payne effect is often referred to as the progressive breakdown of packing-packing interactions, where G’ is used commonly [23,24,25].

RPA8000 was used to conduct three-dimensional strain scanning on CB/NR compounds filled with different contents of OMMT, and the interaction between packing-packing and the interaction between packing-rubber were accurately characterized. Each content was tested with three samples and the experimental results were shown in Figure 3, which indicates that the storage modulus curves of the rubber compound changed significantly with different contents of OMMT. The last two storage modulus curves were greatly lower than the first one under low strain conditions with respect to each different OMMT contents of rubber and the latter two curves were basically coincident. There were packing-packing networks in the composite which was destroyed during the first strain sweep and cannot be instantaneously recovered, and the last two measurements were performed on the basis of the completely destroyed packing-packing network.

In order to further study the dispersion of fillers in the CB/NR composites, we marked it in the RPA scans. The difference values of G’(1)-G’(3) at low strain could indicate the strength of the packing network, and the different values of G’(3)-G’(0) without adding filler at low strain could indicate the strength of the filler and the rubber [26]. It can be seen from Figure 4 that the value of G’(1)-G’(3) with 2 phr OMMT was the lowest in all composites. With adding more OMMT, the value of G’(1)-G’(3) increased progressively. When the content of OMMT was added more than 8 phr, it becomes larger than that of without OMMT. It shows that a small amount (2 and 4 phr) of OMMT induces the weaker filler-filler network, which is helpful for the dispersion of carbon black. If the content is too large, the OMMT agglomeration effect becomes stronger and the material properties decreased. The order of G’(3)-G’(0) value influenced by OMMT was 10 > 8 > 6 > 4 > 0 > 2 phr. When the content of OMMT was 2 phr, the filler-rubber interaction was lowest; the interaction of the filler was also minimized; and the MH-ML was the largest. It can be preliminarily concluded that when 2 phr OMMT added, the rubber-rubber interaction was the strongest and the performance of rubber was good.

Figure 5 shows the storage modulus of the rubber compound filled with different contents of OMMT under small strain, which exhibited a nonlinear decrease with the increase of strain. This was the so-called Payne effect. Normally, the better the fillers dispersion, the lower the storage modulus under the same condition of the strain sweep.

It can be seen from Figure 5 that the rubber composites with 2 phr OMMT had a lowest storage modulus, and the storage modulus was found increase gradually with increasing of OMMT amount. Therefore, a small content of OMMT can improve the dispersibility of filler. In order to analyze the effect of it, ΔG’ (G’0-G’∞) was presented with the variation of OMMT content. It was found that the ΔG’ with 2 phr OMMT is the lowest and more amounts of OMMT made it gradually stronger, which was similar to the results of three consecutive strain scans on it. Since the ΔG’ is analyzed under large strain conditions, the filler network can be completely broken, and its accuracy should be higher than that obtained by the RPA8000 three-strain scan. Based on the results of two strain scans, the order of the filler network strength was obtained as 8 > 6 > 0 > 4 > 2 phr.

As showed in Figure 6, the *T*_g_ of the OMMT/CB/NR composites slightly shifts toward the high temperature direction for the addition amount of 2, 4, and 6 phr OMMT. The slight increase of *T*_g_ is due to the fact that the rubber macromolecular segment is restricted by the sheet structure of OMMT and the exercise capacity is thus reduced. In the case of the 8 phr content, the agglomeration was produced and *T*_g_ was found decrease slightly, which is consistent with the analysis results of RPA. The peak value of Tanδ increased after OMMT was added, which was because the density of cross-linking within the vulcanized rubber increased. In the glass transition zone, the rubber was deformed by the external force and the flaky filler OMMT was oriented subsequently, which would generate friction with the rubber. This will transform the mechanical energy generated by the vibration into the heat energy loss, and then improve the damping of the material. As a result, the damping value of the material increased and the damping temperature range (Tanδ ≥ 0.3) was widened (Figure 7). Finally, the damping performance was increased.

### 3.4. Microstructural Characterization

Figure 8 shows the SEM images of frozen fracture surface (sprayed with gold) of the composites with and without OMMT. The SEM image of the sample without OMMT in Figure 8a shows that the morphology of fractured surface exhibits a relatively smooth surface, except for the presence of small particles of CB. Compared with the surface morphology of the composites sample (0 phr), OMMT particles were occasionally observed on the surface of the composites sample (2 phr OMMT) in the rubber matrix and carbon black, as shown in Figure 8b. Figure 8c,d shows that layered OMMT particles were more to be found in the cross-section of the OMMT/CB/NR composites. With an increase of OMMT in content, the agglomerated OMMT particles were easily observed on the surface of OMMT/CB/NR composite sample (8 phr OMMT) seen in Figure 8e. The lamellar structure of OMMT which may helpful to the damping property of the composite, but the agglomeration phenomenon of OMMT particles was not beneficial to the improvement of mechanical properties and damping performance, as shown in the previous data.

## 4. Conclusions

The CB/NR composite filled with 2 phr OMMT was found to have excellent mechanical properties and was up to the standard of the performance requirements of automobile damping vibration absorbing parts. Compared with the composite without OMMT, the crosslink density of the rubber was increased; the scorch time was extended; and the vulcanization rate was decreased. The OMMT is beneficial to improve the performance of the anti-vibration parts of the automobile and the processing safety. Meanwhile, the addition of OMMT could increase the dispersibility of CB and reduce the final temperature rise of compression heat generation. Both dynamic and static compression permanent deformations were quite small. After filling the OMMT, the system dynamic mechanical and damping properties were enhanced and the effective damping temperature range was broadened.

## Figures and Tables

**Figure 1 polymers-12-01983-f001:**
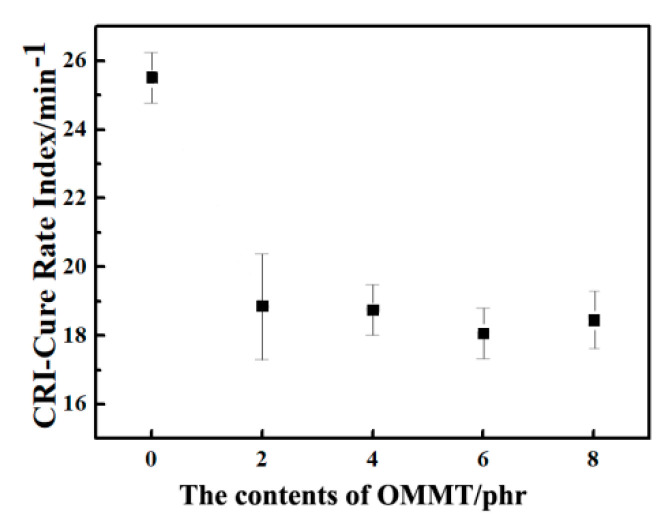
The effects of the contents of OMMT on the cure rate index of carbon black (CB)/natural rubber (NR) composites.

**Figure 2 polymers-12-01983-f002:**
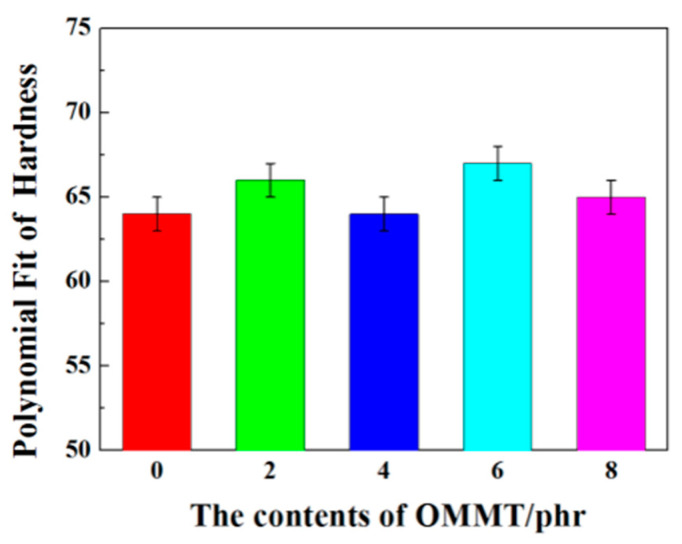
The effects of the contents of OMMT on hardness of vulcanized rubber.

**Figure 3 polymers-12-01983-f003:**
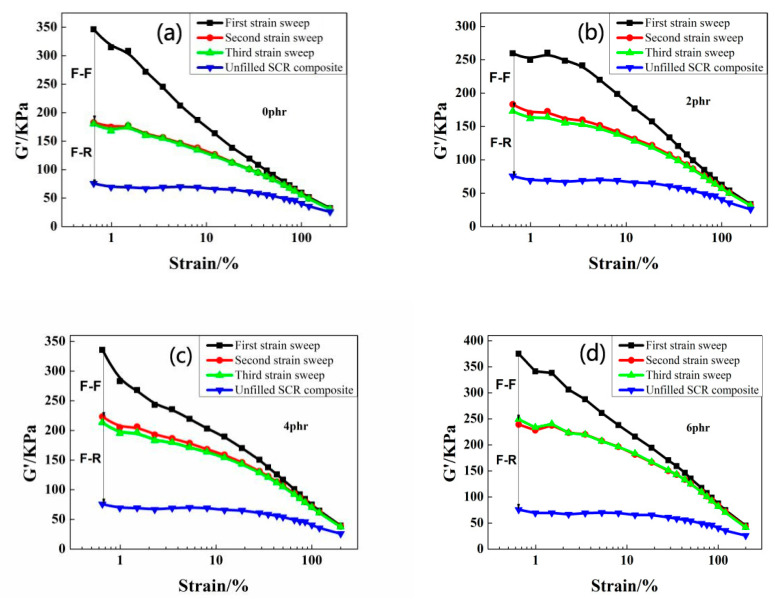
The three times G’-strain curves of the CB/NR rubber composites with different contents of OMMT. The content of OMMT: (**a**) 0 phr, (**b**) 2 phr, (**c**) 4 phr, (**d**) 6 phr, (**e**) 8 phr.

**Figure 4 polymers-12-01983-f004:**
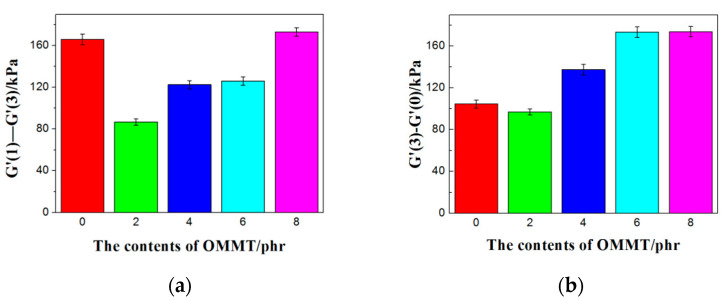
(**a**) G’ (1)-G’ (3) and (**b**) G’ (3)-G’ (0) of CB/NR composites with different contents of OMMT.

**Figure 5 polymers-12-01983-f005:**
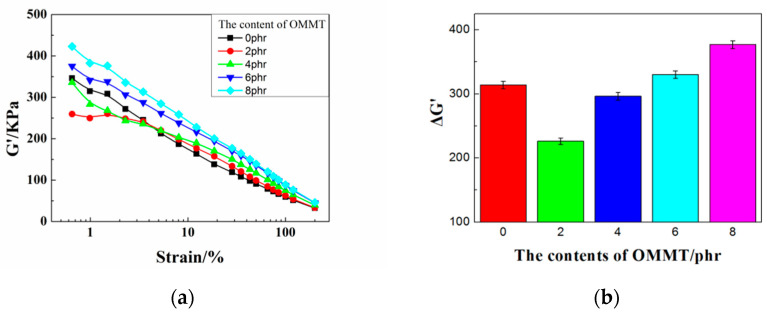
(**a**) G’-strain and (**b**) ΔG’ of CB/NR composites with different contents of OMMT.

**Figure 6 polymers-12-01983-f006:**
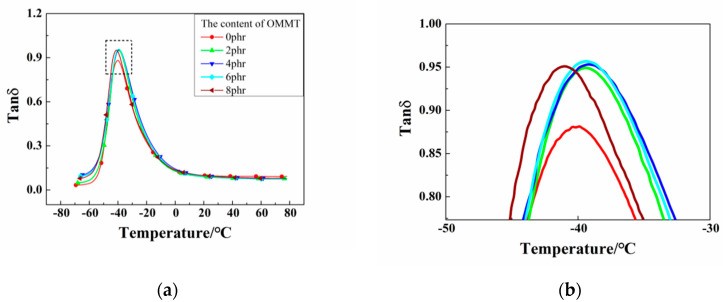
The loss factor-temperature curves of the CB/NR composites with different contents of OMMT. (Note: (**b**) is a local enlarged view of (**a**).)

**Figure 7 polymers-12-01983-f007:**
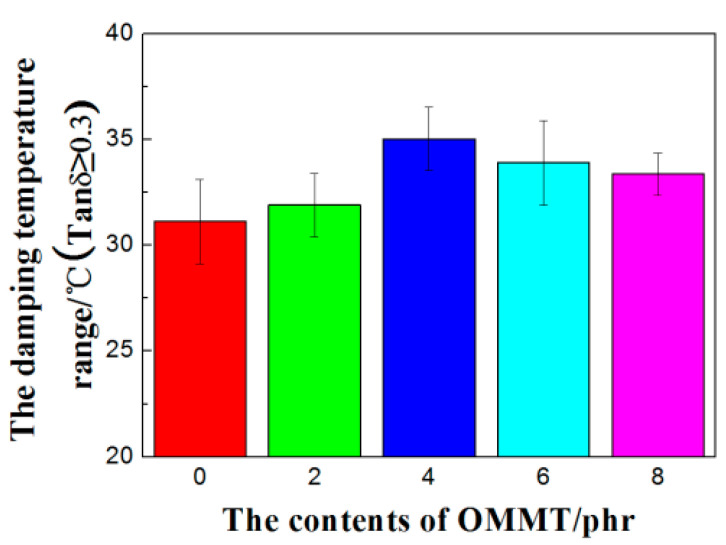
Comparison the damping temperature of CB/NR with different contents of OMMT.

**Figure 8 polymers-12-01983-f008:**
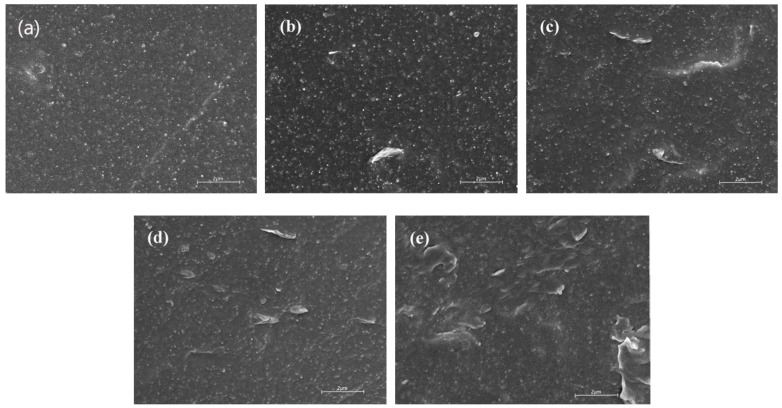
SEM images of rubber sections with various OMMT contents: (**a**) 0 phr, (**b**) 2 phr, (**c**) 4 phr, (**d**) 6 phr, and (**e**) 8 phr.

**Table 1 polymers-12-01983-t001:** Formula of natural rubber (NR) with organic montmorillonite (OMMT).

Ingredient	NR	Sulfur	ZnO	Stearic Acid	Microcrystalline Wax	CTP	4020	RD	N330	OMMT
Phr	100	2	5	1	1	1	3	1	50	0/2/4/6/8

**Table 2 polymers-12-01983-t002:** The effects of OMMT contents on mechanical properties of vulcanized rubber.

The Content of OMMT/phr	0	2	4	6	8
Tensile Strength/MPa	27.5	27.6	25.2	25.6	24.4
Elongation/%	616	502	519	515	488
Tensile Strength at 100%/MPa	2.1	3.2	2.7	2.8	2.8
Tensile Strength at 300%/MPa	11.0	15.0	12.4	12.9	13.2
Elasticity Modulus/MPa	2.6	3.4	2.9	3.1	3.2
Tear Strength/kN/m	131.9	111.9	111.3	96.6	87.0
Resilience Property/%	47.7	51.9	51.4	52.0	51.4

**Table 3 polymers-12-01983-t003:** The effects of OMMT content on the compressed heat generation and compression permanent deformation of vulcanized rubber.

OMMT/phr	Final Temperature Rise/°C	Dynamic Compression Permanent Deformation/%	Static Compression Permanent Deformation/%
0	15.85	0.060	0.093
2	13.90	0.040	0.091
4	14.25	0.051	0.083
6	15.30	0.055	0.081
8	15.80	0.045	0.072

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
