# Peer review of "The Effect of OMMT on the Properties of Vehicle Damping Carbon Black-Natural Rubber Composites"

_polymers, 2020, doi:10.3390/polym12091983_

Round 1
Reviewer 1 Report
In my opinion, the article: "The Effect of OMMT on the Properties of Vehicle Damping Carbon Black-Natural Rubber Composites" deserves to be published. The main issues and critical aspect found in the previous versions have been properly corrected.
Author Response
Manuscript ID: Polymers-888782
Title: " The Effect of OMMT on the Properties of Vehicle Damping Carbon Black-Natural Rubber Composites "
Author(s): Liu Wei , Lv Lutao , Yang Zonglin , Zheng Yuqing , Hui Wang
First of all, I would like to thank you for giving us the opportunity to revise this manuscript. We also appreciate the two reviewers’ comments which help us to improve the research quality.
We have carefully revised our manuscript according to the suggestions of both reviewers. All revisions are high-lighted in the revised manuscript.
Reviewer #1:
The following modifications have been made to the article:
- When 5 phr graphite was added to nitrile rubber to replace the same weight of carbon black,the storage modulus of nitrile rubber increased correspondingly, and the effective damping temperature range expanded, which are beneficial to improve the damping performance.
(Lines 41-44)
- In this study, the damping properties of rubber composites were improved from the perspective of structural packing.(Lines 68-69)
(3)- "Noor Azammi [15] filled kenaf fiber into TPU-NR composites and got a sample that would shift the damping temperature range up to 135℃" (Lines 45-46)
(4) When OMMT was used into rubber, the MH-ML of compound increased (Lines 138-139)
- The type of ZnO and other indetail description were added In the raw materials section.
Natural rubber (SCR, Hainan Natural Rubber Company’s product) was used as basis material. Sulfur (99.5% purity) was purchased from Jinchangsheng Co., China. Organic montmorillonite (OMMT, 85%, organic modified by quaternary ammonium salt, particle size 44 μm, colloidal viscosity 4.0 mPas•s) was purchased from Zhejiang Fenghong clay chemical Co., China. Cabon Black (N330) was purchased from Cabot Corporation. N-cyclohexyl thio-phthalimide (CTP-70GR, a kind of scorch retarder), N-1,3-dimethylbutyl-N'-phenyl-p-phenylenediamine (4020, a kind of antioxidant), Poly (1,2-dihydro-2,2,4-trimethyl-quinoline) (RD, a kind of antioxidant) and Zinc oxide (ZnO-80) were purchased from Ningbo Actmix Rubber Chemicals Co., Ltd. The remaining ingredients were provided by Jinchangsheng Co., China. Table 1 shows the specific formula proportion of samples.(line78-87);
(6) Fixed the missing apaces before references in line 27, 28, 33, 36, 47, 52, 55, 63 and 206. The formatting in line 38 changed as [8-12] .
(7)I had look for all the tables and figures, and replaced them with TABLE or FIGURE.
(8)The font size in plots have been changed, and all the Chinese symbols have erased.
FIGURE 4. G' (1)-G' (3) and G' (3)-G' (0) of CB/NR composites with different contents of OMMT.
(9)In this study, our work used OMMT to improve the damping performance of rubber materials, and the ideal result is not the best with the state of the art , but this work have some novelty and reference value, we think.
(10)For the preparation procedures, all the materials of the formula listed in table 1 were weighed proportionally, the mass of NR is 200g. That means that NR has a mass of 200 grams.
Appropriate numbering have been added from 2.2.1 to 2.2.5.
(12) The error bars are close but not the same. They were determined by actual data from multiple trials conducted in accordance with the test criteria.
(13)Lines between the symbols have been deleted, and the curve was not polynomial fitting curve, I deleted the word “polynomial”.
(14) This is really a cross-section image of the SEM micrographs, its a frozen fracture surface (sprayed with gold) of the composites. In SEM images, we can not support the composite is homogeneous, so the sentence is changed as:
Figure 8 shows the SEM images of frozen fracture surface (sprayed with gold) of the composites with and without OMMT. SEM image of the sample without OMMT in figure 8(a) showed that the morphology of fractured surface exhibits a relatively smooth surface, except for the presence of small particles of CB. Compared with the surface morphology of the composites sample (0 phr), OMMT particles were occasionally observed on the surface of the composites sample (2 phr OMMT) in the rubber matrix and carbon black, as shown in figure 8(b). Figure 8(c) and (d) showed that layered OMMT particles were more to be found in the cross-section of the OMMT/CB/NR composites. With an increase of OMMT in content, the agglomerated OMMT particles were easily observed on the surface of OMMT/CB/NR composite sample (8 phr OMMT) seen in figure 8(e).
Thank you very much for all your comments!

Reviewer 2 Report
This submission authored by Liu and co-workers demonstrated the synthesis of rubber composites and characterization of their mechanical properties. In the present state, it cannot be accepted by the journal, but the incorporation of the correction suggestions may make it appropriate for publication in Polymers.
1) The work should be proofread by a native English speaker. The manuscript contains numerous errors, which make it nearly impossible to understand the concept. For instance:
- "When 5 phr graphite added into nitrile rubber instead of carbon black, the storage modulus of the nitrile rubber material was increased accordingly, and the effective damping temperature range became widely, which both were beneficial to improve the damping performance" (Lines 41-43)
- "In this research, in order to improve the damping performance of rubber composites from the angle of structural packing" (Lines 67 and 68) - in order to what?
- "Noor Azammi [15] filled nature fibers into TPU-NR composites and got a sample that would shift the damping temperature range up to 135℃" (Lines 44-46) - what are nature fibers?
- "When the filler OMMT was used into rubber, the MH-ML of compound increased" (Lines 136-137)
- "Then, the CTP(N-cyclohexyl thio88 phthalimide, a kind of scorch retarder), 4020(N-1,3-dimethylbutyl-N'-phenyl-p-phenylenediamine, a kind of antioxidant), RD(poly(1,2-dihydro-2,2,4-trimethyl-quinoline), a kind of antioxidant), mixed for 3 minutes. The follwing N330(CB) and OMMT, for 3 minutes. The last, sulfur and ZnO (Zinc oxide) were added into the blended compound" - many errors and lacks clarity. Moreover, what type of ZnO is used?
Etc. There are more of them, so please locate all the errors and fix them.
2) The formatting is also wrong in many places. Examples of errors include things like:
- [8~12] (Line 38)
- missing spaces before references
- tables or figures should have capital letters (example of a mistake, Line 83). Look for all of them and make appropriate corrections.
- font size in plots is too small and at 100% magnification it is not possible to read the figures
- there are some Chinese symbols in Fig. 4a
3) Please compare the results of this study with the state of the art. At present, it is not possible to determine whether this submission presents any improvement.
4) The description below Table 1. Please correct it as it is very confusing. In addition, are these proportions presented on the basis of weight or volume? Lastly, you use various amounts of OMMT: 0, 2, 4, 6, and 8. You also specify that these proportions are provided for 200 g total weight. Therefore, I would like to ask you what is added to keep the weight constant as different amount of OMMT is used.
6) Section 2.2. These headlines such as Mechanical Tests and RPA should take appropriate numbering such as 2.2.X provided by the MDPI template.
7) Please describe how statistics were determined for these data. I am interested to find out why error bars in Fig. 1 and Fig. 2 appear the same for all the content values of OMMT.
8) Fig. 2 - lines between the symbols are not justified since you did not measure values in between. You should keep the symbols, but leave out the black line. Regarding the polynomial fit - why polynomial?
9) Description related to micrographs - "Compared with the surface morphology of the composites sample (0 phr), OMMT particles were observed on the surface of the composites sample (2 phr OMMT) and separately dispersed in the rubber matrix and carbon black, as shown in Figure 8(b). Figure 8(d) and (e) showed that layered OMMT particles were still uniformly dispersed in the cross section of the OMMT/CB/NR composites. With an increase of OMMT in content, the agglomerated OMMT particles were easily observed on the surface of OMMT/CB/NR composite sample (8 phr OMMT) seen in Figure 8(f)." (Lines 248-254).
- What are these circles? What do they indicate?
- You claim that the composite is homogeneous, but the images do not support this hypothesis.
- Is this really a cross-section image of the SEM micrographs of the top of the sample?
- Were these samples sputtered with metal for analysis?
Author Response
Manuscript ID: Polymers-888782
Title: " The Effect of OMMT on the Properties of Vehicle Damping Carbon Black-Natural Rubber Composites "
Author(s): Liu Wei , Lv Lutao , Yang Zonglin , Zheng Yuqing , Hui Wang
First of all, I would like to thank you for giving us the opportunity to revise this manuscript. We also appreciate the two reviewers’ comments which help us to improve the research quality.
We have carefully revised our manuscript according to the suggestions of both reviewers. All revisions are high-lighted in the revised manuscript.
Reviewer #2:
This submission authored by Liu and co-workers demonstrated the synthesis of rubber composites and characterization of their mechanical properties. In the present state, it cannot be accepted by the journal, but the incorporation of the correction suggestions may make it appropriate for publication in Polymers.
1) The work should be proofread by a native English speaker. The manuscript contains numerous errors, which make it nearly impossible to understand the concept. For instance:
- "When 5 phr graphite added into nitrile rubber instead of carbon black, the storage modulus of the nitrile rubber material was increased accordingly, and the effective damping temperature range became widely, which both were beneficial to improve the damping performance" (Lines 41-43)
Reply:
When 5 phr graphite was added to nitrile rubber to replace the same weight of carbon black, the storage modulus of nitrile rubber increased correspondingly, and the effective damping temperature range expanded, which are beneficial to improve the damping performance.
(Lines 41-44)
- "In this research, in order to improve the damping performance of rubber composites from the angle of structural packing" (Lines 67 and 68) - in order to what?
Reply:
In this study, the damping properties of rubber composites were improved from the perspective of structural packing.(Lines 68 -69)
- "Noor Azammi [15] filled nature fibers into TPU-NR composites and got a sample that would shift the damping temperature range up to 135℃" (Lines 44-46) - what are nature fibers?
Reply:
- "Noor Azammi [15] filled kenaf fiber into TPU-NR composites and got a sample that would shift the damping temperature range up to 135℃" (Lines 45-46)
- "When the filler OMMT was used into rubber, the MH-ML of compound increased" (Lines 136-137)
Reply:
When OMMT was used into rubber, the MH-ML of compound increased (Lines 138-139)
- "Then, the CTP(N-cyclohexyl thio88 phthalimide, a kind of scorch retarder), 4020(N-1,3-dimethylbutyl-N'-phenyl-p-phenylenediamine, a kind of antioxidant), RD(poly(1,2-dihydro-2,2,4-trimethyl-quinoline), a kind of antioxidant), mixed for 3 minutes. The follwing N330(CB) and OMMT, for 3 minutes. The last, sulfur and ZnO (Zinc oxide) were added into the blended compound" - many errors and lacks clarity. Moreover, what type of ZnO is used?
Etc. There are more of them, so please locate all the errors and fix them.
Reply:
The type of ZnO and other indetail description were added In the raw materials section.
Natural rubber (SCR, Hainan Natural Rubber Company’s product) was used as basis material. Sulfur (99.5% purity) was purchased from Jinchangsheng Co., China. Organic montmorillonite (OMMT, 85%, organic modified by quaternary ammonium salt, particle size 44 μm, colloidal viscosity 4.0 mPas•s) was purchased from Zhejiang Fenghong clay chemical Co., China. Cabon Black (N330) was purchased from Cabot Corporation. N-cyclohexyl thio-phthalimide (CTP-70GR, a kind of scorch retarder), N-1,3-dimethylbutyl-N'-phenyl-p-phenylenediamine (4020, a kind of antioxidant), Poly (1,2-dihydro-2,2,4-trimethyl-quinoline) (RD, a kind of antioxidant) and Zinc oxide (ZnO-80) were purchased from Ningbo Actmix Rubber Chemicals Co., Ltd. The remaining ingredients were provided by Jinchangsheng Co., China. Table 1 shows the specific formula proportion of samples.(line78-87);
2) The formatting is also wrong in many places. Examples of errors include things like: - [8~12] (Line 38)- missing spaces before references
Reply:
Fixed the missing apaces before references in line 27, 28, 33, 36, 47, 52, 55, 63 and 206. The formatting in line 38 changed as [8-12] .
tables or figures should have capital letters (example of a mistake, Line 83). Look for all of them and make appropriate corrections.
Reply:
I had look for all the tables and figures, and replaced them with TABLE or FIGURE.
 - font size in plots is too small and at 100% magnification it is not possible to read the figures
- there are some Chinese symbols in Fig. 4a
Reply:
The font size in plots have been changed, and all the Chinese symbols have erased.
FIGURE 4. G' (1)-G' (3) and G' (3)-G' (0) of CB/NR composites with different contents of OMMT.
3) Please compare the results of this study with the state of the art. At present, it is not possible to determine whether this submission presents any improvement.
Reply:
In this study, our work used OMMT to improve the damping performance of rubber materials, and the ideal result is not the best with the state of the art , but this work have some novelty and reference value, we think.
4) The description below Table 1. Please correct it as it is very confusing. In addition, are these proportions presented on the basis of weight or volume? Lastly, you use various amounts of OMMT: 0, 2, 4, 6, and 8. You also specify that these proportions are provided for 200 g total weight. Therefore, I would like to ask you what is added to keep the weight constant as different amount of OMMT is used.
Reply:
For the preparation procedures, all the materials of the formula listed in table 1 were weighed proportionally, the mass of NR is 200g. That means that NR has a mass of 200 grams.
6) Section 2.2. These headlines such as Mechanical Tests and RPA should take appropriate numbering such as 2.2.X provided by the MDPI template.
Reply:
Appropriate numbering have been added from 2.2.1 to 2.2.5.
7) Please describe how statistics were determined for these data. I am interested to find out why error bars in Fig. 1 and Fig. 2 appear the same for all the content values of OMMT.
Reply:
The error bars are close but not the same. They were determined by actual data from multiple trials conducted in accordance with the test criteria.
8) Fig. 2 - lines between the symbols are not justified since you did not measure values in between. You should keep the symbols, but leave out the black line. Regarding the polynomial fit - why polynomial?
Reply:
Lines between the symbols have been deleted, and the curve was not polynomial fitting curve, I deleted the word “polynomial”.
9) Description related to micrographs - "Compared with the surface morphology of the composites sample (0 phr), OMMT particles were observed on the surface of the composites sample (2 phr OMMT) and separately dispersed in the rubber matrix and carbon black, as shown in Figure 8(b). Figure 8(d) and (e) showed that layered OMMT particles were still uniformly dispersed in the cross section of the OMMT/CB/NR composites. With an increase of OMMT in content, the agglomerated OMMT particles were easily observed on the surface of OMMT/CB/NR composite sample (8 phr OMMT) seen in Figure 8(f)." (Lines 248-254).
- What are these circles? What do they indicate?
- You claim that the composite is homogeneous, but the images do not support this hypothesis.
- Is this really a cross-section image of the SEM micrographs of the top of the sample?
- Were these samples sputtered with metal for analysis?
Reply:
-As you may have wondered, circles don't make any sense in figure 8 , so I deleted them.
This is really a cross-section image of the SEM micrographs, its a frozen fracture surface (sprayed with gold) of the composites. In SEM images, we can not support the composite is homogeneous, so the sentence is changed as:
Figure 8 shows the SEM images of frozen fracture surface (sprayed with gold) of the composites with and without OMMT. SEM image of the sample without OMMT in figure 8(a) showed that the morphology of fractured surface exhibits a relatively smooth surface, except for the presence of small particles of CB. Compared with the surface morphology of the composites sample (0 phr), OMMT particles were occasionally observed on the surface of the composites sample (2 phr OMMT) in the rubber matrix and carbon black, as shown in figure 8(b). Figure 8(c) and (d) showed that layered OMMT particles were more to be found in the cross-section of the OMMT/CB/NR composites. With an increase of OMMT in content, the agglomerated OMMT particles were easily observed on the surface of OMMT/CB/NR composite sample (8 phr OMMT) seen in figure 8(e).
Thank you very much for all your comments!

Round 2
Reviewer 2 Report
The article has been improved, but some of the comments have been neglected.
1) Please compare the results of this study with the state of the art. At present, it is not possible to determine whether this submission presents any improvement.
Reply of the authors:
In this study, our work used OMMT to improve the damping performance of rubber materials, and the ideal result is not the best with the state of the art , but this work have some novelty and reference value, we think.
Reply of the reviewer:
That is why it is essential that you provide a precise statement in the article what is novel and valuable about this work. It is a standard practice in scientific papers.
2) The description below Table 1. Please correct it as it is very confusing. In addition, are these proportions presented on the basis of weight or volume? Lastly, you use various amounts of OMMT: 0, 2, 4, 6, and 8. You also specify that these proportions are provided for 200 g total weight. Therefore, I would like to ask you what is added to keep the weight constant as different amount of OMMT is used.
Reply of the authors:
For the preparation procedures, all the materials of the formula listed in table 1 were weighed proportionally, the mass of NR is 200g. That means that NR has a mass of 200 grams.
Reply of the reviewer:
Please answer my questions.
3) Please describe how statistics were determined for these data. I am interested to find out why error bars in Fig. 1 and Fig. 2 appear the same for all the content values of OMMT.
Reply of the authors:
The error bars are close but not the same. They were determined by actual data from multiple trials conducted in accordance with the test criteria.
Reply of the reviewer:
Be more specific to attain the necessary level of rigor. For example, how many samples were tested.
4) Fig. 2 - lines between the symbols are not justified since you did not measure values in between. You should keep the symbols, but leave out the black line. Regarding the polynomial fit - why polynomial?
Reply of the authors:
Lines between the symbols have been deleted, and the curve was not polynomial fitting curve, I deleted the word “polynomial”.
Reply of the reviewer:
Figure 1 still contains a line between the measurement point. You also disregarded my question why the fit is polynomial. It is not enough to remove the word from the plot. The readers need to know why exactly such a curve shape was used for fitting.
5) There are also some minor mistakes. For instance, figure captions should not have capitalized words "FIGURE", but should read "Figure" instead.
Author Response
Manuscript ID: Polymers-888782
Title: " The Effect of OMMT on the Properties of Vehicle Damping Carbon Black-Natural Rubber Composites "
Author(s): Liu Wei , Lv Lutao , Yang Zonglin , Zheng Yuqing , Hui Wang
Thank you very much for giving us the opportunity again to revise this manuscript. Also we appreciate the reviewers’ comments which help us to improve the research quality.
All revisions are high-lighted in the revised manuscript.
Reviewer :
The article has been improved, but some of the comments have been neglected.
1) Please compare the results of this study with the state of the art. At present, it is not possible to determine whether this submission presents any improvement.
Relpy:
It is rarely reported that OMMT was used to improve the vibration damping performance of rubber, while the layered structure of it is worth studying. At the same time, it can be found from the previous research work that the damping performance of rubber is mainly affected by the structure of rubber and its filler. In this study, the processing performance, mechanical properties and damping properties of rubber composites were all improved only with little amount of OMMT.
2) The description below Table 1. Please correct it as it is very confusing. In addition, are these proportions presented on the basis of weight or volume? Lastly, you use various amounts of OMMT: 0, 2, 4, 6, and 8. You also specify that these proportions are provided for 200 g total weight. Therefore, I would like to ask you what is added to keep the weight constant as different amount of OMMT is used.
Relpy: I apologize for my unclear solution.
For the first question, The description below Table 1 is changed :
The formula of this study was listed in Table 1, on the basis of weight. In the preparation process, all raw materials were weighed according to the formula ratio, and the actual weight of NR was 200g. Firstly, NR was dried at 60℃ for 4h before use. Then, it was processed with stearic acid and microcrystalline wax in sk-160B two-roll mill (Shanghai Plastics and Rubber Machinery Co., China) at room temperature, the initial distance between the two rollers was 2mm, completely encased on the roller for 2 minutes. Next, the CTP (N-cyclohexyl thio-phthalimide, a kind of scorch retarder), 4020 (N-1,3-dimethylbutyl-N'-phenyl-p-phenylenediamine, a kind of antioxidant), RD (poly (1,2-dihydro-2,2,4-trimethyl-quinoline), a kind of antioxidant), mixed for 3 minutes. The following were carbon black N330 and OMMT, mixed for 3 minutes. After that, sulfur and ZnO (Zinc oxide) were added into the blended compound. Next, set the distance between the two rollers to 1mm, made the "triangle bag" in the process for 5 times. At last, the compound obtained was molded into sheets in a Type Plate Vulcanizer (XLB-400X400X2 50T) (Qingdao Yadong Machinery, China) at 150 ℃ for 15 min.
For the second question,
Table 2 shows the effect of OMMT addition amount (on the basis of weight) on the mechanical properties of vulcanized CB/NR polymer composites. --it is on the basis of weight.
Last:
Customarily ,in the research of rubber material performance, the influence of different packing amount on rubber performance was investigated based on 100 phr rubber. So, it can not keep the weight constant as different amount of OMMT is used.
3) Please describe how statistics were determined for these data. I am interested to find out why error bars in Fig. 1 and Fig. 2 appear the same for all the content values of OMMT.
Relpy: Mechanical properties were used to tested with 5 samples, 3 samples for RPA, DMA and Processing Performance, and 1 sample for SEM. It is explained in the experiment section.( section 2.2.1, 2.2.2, 2.2.3, 2.2.4, 2.2.5 )
4) Fig. 2 - lines between the symbols are not justified since you did not measure values in between. You should keep the symbols, but leave out the black line. Regarding the polynomial fit - why polynomial?
Relpy: As you have discussed, the black line between the measurement points should be left, and the polynomial fit are not justified since I did not measure values in between. So, The related statement is changed to:”Figure 2 shows the hardness of composite material changing with the amounts of OMMT. It can be seen that the hardness values of vulcanized rubber filled different contents of OMMT were all within 65±2 and the general trend of it is increase slightly with more contents, but not all of it fits this pattern due to experimental mistakes. It also reflects its effect on the packing-packing network of vulcanized rubber. These data indicates that OMMT has minor influence on the vulcanized rubber hardness, which is of great importance to the formulation design of shock absorbing products.” (from Line 160 to 166)
And the Figure 1 and Figure 2 were also changed.
- There are also some minor mistakes. For instance, figure captions should not have capitalized words "FIGURE", but should read "Figure" instead.
Relpy: All the “figure” and “table” in the manuscript have been searched and read as “Figure” or “Table” instead.

Round 3
Reviewer 2 Report
The article can be accepted in the present form.
This manuscript is a resubmission of an earlier submission. The following is a list of the peer review reports and author responses from that submission.
Round 1
Reviewer 1 Report
In this manuscript by Liu et al. the authors have investigated the properties of carbon black-rubber composites. The work is focuced on evaluation of the damping characteristics of the material. It may be reconsidered by Polymers upon incorporation of the following suggestions.
1) State of the art regarding OMMT and rubber is not precisely defined. It is doubtful that the three provided sentences summarize the whole area. Please expand it.
2) Furthermore, the "in this research [..]" section in the introduction should be more detailed.
3) The manuscript is not formatted well - The references are not in line with MDPI requirements, panels of the schemes should be assigned in the topleft corner not on the bottom, etc. Please proofread it paying necessary attention to English because there are errors which make it very hard to understand the concept.
4) phr and other acronyms are not defined.
5) "The remaining ingredients were commercially available. " - provide the source
6) The description of processing in 2.1 is not rigorous enough. The primary aim of a scientific paper is to enable others to reproduce the study and if there are no details then it will be impossible. Please make the description more specific. The same remark is valid to other sections in the experimental part, which also lack detail.
7) No statistics. How many samples were tested for each parameter set? If one, the drawn conclusions may be misleading.
8) SEM is required to support many claims of this article.
Author Response
Response to Reviewer 1 Comments
The difference between our article Polymers-new and Polymers-853891 is as follows:
Your valuable comments:
Response 1: State of the art regarding OMMT and rubber is defined more:
Add the The Summary of previous work from the references [1],[2],[14],[15],[19],[21],[22]and[23].
Response 2:Furthermore, the "in this research [..]" section, Some modifications have been made and the introduction had be detailed and .
Response 3: Format modification in line with MDPI requirements:
(1) The format of references have been modified as required.
(2) The panels of the schemes have been assigned in the topleft corner.
Response 4 and 5: Some acronyms and some sources are defined:
(1) Phr: parts per hundreds of rubber
(2)CTP:N-cyclohexyl thio-phthalimide, a kind of scorch retarder
(3)4020:N-1,3-dimethylbutyl-N'-phenyl-p-phenylenediamine, a kind of antioxidant
(4)RD:poly(1,2-dihydro-2,2,4-trimethyl-quinoline), a kind of antioxidant
(5)N330:CB
(6)ZnO:Zine oxide
(7)The remaining ingredients were provided by Jinchangsheng Co., China.
Response 6: Additional informations have been described:
Make the description of the distance of roller and the mixing time for each processing in 2.1.
Response 7:The samples tested in the work are all following each experimental standards, so there were no more explainations for them.
Response 8:SEM is added to support the micromorphology and the dispersion of OMMT in each composite sample.
Other modifications:
- Re-write the sentences which were not readble:
(1) Pag. 1. raws 41-42: "...which make the effective damping temperature range widely". modified to :
“ When 5 phr graphite added into nitrile rubber instead of carbon black, the storage modulus of the nitrile rubber material was increased accordingly, and the effective damping temperature range became widely, which both were beneficial to improved the damping performance. ”
(2)Pag. 2. - raws 48-49: "...and the other performances almost kept 48 well." modified to :
“...the mechanical properties and high-temperature performances almost kept well”
- 2 - raws 52-53: "... These features have made OMMT become a new rubber reinforcing filler" modified to :
As a result, it is used as a new reinforcing filler in rubber.
- For the Questions about Table 2:
The modulus can be concluded by the tensile strength at different elongations, so it was modified to :
According to the tensile strength at different elongations, it can be seen that the modulus had a decreasing trend with increasing of the OMMT contents.
- For the Questions about table 3:
It had been modified according to the opinions.
- For the Questions about pages 6. - raw 171,
I think it would be used the term “weaker” here.

Reviewer 2 Report
In my opinion the article "The Effect of OMMT on the Properties of Vehicle Damping Carbon Black-Natural Rubber Composites", prior to be published needs a deep revision in the English language (mother tongue is suggested). In general, from a scientific point of view, the article is quite interesting, the experimental part has been correctly and reasonably carried out; results are very good and the conclusions can be agreed, but in many point the authors message can't be completely understood just because of the English language.
Here some other examples:
Pag. 1. raws 41-42: The authors claim: "...which make the effective damping temperature range widely".
- The authors are invited to completely re-write the sentence, as it is not readable. Moreover, the English language has to be improved in the whole page, despite the authors message can be argued at least;
· Pag. 2. - raws 48-49. The authors claims: "...and the other performances almost kept 48 well."
- The authors are invited to completely re-write the sentence, as it is not readable;
· Pag. 2 - raws 52-53. The authors claims: "... These features have made OMMT become a new rubber reinforcing filler"
- The authors are invited to completely re-write the sentence, as it is not readable and also the authors message can't be argued;
· Table 2. In the paragraph the topic of the systems modules is dealt with, but this properties is not summarized in the table. In my opinion, the modulus values for any system have to be added.
· Pages 4 and 5. - the table 3 caption has to be written in the same page ha the corresponding table.
- The authors are invited to improve the manuscript layout;
· Pages 6. - raw 171. probably the term "weaker" has to be replaced with "weakest";
Author Response
Response to Reviewer 2 Comments(Please see the attachment for the new manuscripts . )
The difference between our article Polymers-new and Polymers-853891 is as follows:
Your valuable comments:
Response 1-3: Re-write the sentences which were not readble:
(1) Pag. 1. raws 41-42: "...which make the effective damping temperature range widely". modified to :
“ When 5 phr graphite added into nitrile rubber instead of carbon black, the storage modulus of the nitrile rubber material was increased accordingly, and the effective damping temperature range became widely, which both were beneficial to improved the damping performance. ”
(2)Pag. 2. - raws 48-49: "...and the other performances almost kept 48 well." modified to :
“...the mechanical properties and high-temperature performances almost kept well”
- 2 - raws 52-53: "... These features have made OMMT become a new rubber reinforcing filler" modified to :
As a result, it is used as a new reinforcing filler in rubber.
Response 4: For the Questions about Table 2:
The modulus can be concluded by the tensile strength at different elongations, so it was modified to :
According to the tensile strength at different elongations, it can be seen that the modulus had a decreasing trend with increasing of the OMMT contents.
Response 5: For the Questions about table 3:
It had been modified according to the opinions.
Response 6: For the Questions about pages 6. - raw 171,
I think it would be used the term “weaker” here.
Other modifications:
- State of the art regarding OMMT and rubber is defined more:
Add the The Summary of previous work from the references [1],[2],[14],[15],[19],[21],[22]and[23].
- Furthermore, the "in this research [..]" section, Some modifications have been made and the introduction had be detailed and .
- Format modification in line with MDPI requirements:
(1) The format of references have been modified as required.
(2) The panels of the schemes have been assigned in the topleft corner.
- Some acronyms and some sources are defined:
(1) Phr: parts per hundreds of rubber
(2)CTP:N-cyclohexyl thio-phthalimide, a kind of scorch retarder
(3)4020:N-1,3-dimethylbutyl-N'-phenyl-p-phenylenediamine, a kind of antioxidant
(4)RD:poly(1,2-dihydro-2,2,4-trimethyl-quinoline), a kind of antioxidant
(5)N330:CB
(6)ZnO:Zine oxide
(7)The remaining ingredients were provided by Jinchangsheng Co., China.
- Additional informations have been described:
Make the description of the distance of roller and the mixing time for each processing in 2.1.
- The samples tested in the work are all following each experimental standards, so there were no more explainations for them.
- SEM is added to support the micromorphology and the dispersion of OMMT in each composite sample.

Round 2
Reviewer 1 Report
Thank you for providing the revised version. It would be better if all the pages contained line numbering on the left (the reviewer could then specify places in the manuscript, which require further work). Some improvements have been made, but certain issues have not been taken care of enough:
1) It is still not clear if this study gives any benefits as compared with the state of the art. It was only mentioned that some researchers did a similar study to elucidate the mechanism of damping. How about you compare the performance?
2) "In this research" section lacks purpose. Add at least one sentence summarizing why this article is important.
3) Letters describing the panels are barely visible. It would be useful to put them in bold. Moreover, it is better to place them in the top left corner as I suggested because that is where one would expect to find these.
4) Despite my remark to correct English, it was disregarded. There are many parts, which are completely illegible. For instance:
- "Noor Azammi [15] filled nature fibers into TPU-NR composites and got a sample which had good damping properties in the high-temperature condition up to 135℃" (it would be useful to give a quantitative description of what is "good"). This is a manuscript for publication not a laboratory report.
- "Xu et al.[20] prepared a rubber composite with excellent properties by using multilayered structure material
and reveal the damping mechanism of revealed the damping mechanism of the at the molecular level and in a quantitative manner" (in this case readers will not have any clue what you mean by excellent)
- "The micromorphology of OMMT in rubber matrix were observed using a JEOL JSM-7500F scanning electron microscope (Japanese electronics company, Japanese)."
- " ZnO (Zine oxide)" - should read Zinc oxide
There are countless errors like these.
5) My comments regarding the fact that the experimental section is vague have been disregarded.
"For the preparation procedures, the NR was firstly dried at 60℃ for 4h before use. It was processed at room temperature with stearic acid and microcrystalline wax on an SK-160B two-roll mill (Shanghai Plastics and Rubber Machinery Co., China).), the initial distance between the two mills was 2mm, banding the roll for 2 minutes. Then, the CTP, (N-cyclohexyl thio-phthalimide, a kind of scorch retarder), 4020,(N-1,3-dimethylbutyl-N'-phenyl-p-phenylenediamine, a kind of antioxidant),RD, (poly(1,2-dihydro-2,2,4-trimethyl-quinoline), a kind of antioxidant), mixed for 3 minutes. The follwing N330 ,(CB) and OMMT, for 3 minutes. The last, sulfur and ZnO (Zine oxide) were added in order into the blended compound. Next, set the distance between the two mills to 1mm, made triangle bag and roll 5 times respectively. After that, the compound obtained was molded into sheets in a Type Plate Vulcanizer (XLB-400X400X2 50T) (Qingdao Yadong Machinery, China) at 150 ℃ for 15 min. " - nobody knows in what ratios these compounds are combined. I admit that there is Table 1 above, but it is not specified whether these proportions are provided on weight basis.
6) Regarding the issue with the lack of statistics, your answer is unclear "The samples tested in the work are all following each experimental standards, so there were no more explainations for them". From what I can see, Fig. 1 (no error bars), Fig. 4 (no error bars), Fig. 7 (no error bars). I tried to refer to them correctly, but I think that the current numbering is not correct.
7) SEM analysis shows that distribution is not even, so I would disagree with "Compared with the surface morphology of OMMT/CB/NR composite sample (0 phr), OMMT particles were observed on the surface of the OMMT/CB/NR composite sample (2 phr OMMT) and their distribution was relatively even, as shown in Figure 8(b)."
Besides, SEM micrographs do not have professional scale bar markers.
Reviewer 2 Report
In my opinion, the article " The Effect of OMMT on the Properties of Vehicle Damping Carbon Black-Natural Rubber Composites" deserves to be published after a minor revision. At this purpose, despite the English language has been significantly improved, some sections still remain characterised by an inadequate readability.
Here some question:
· Abstract: replace "nature rubber" with "natural rubber";
· Pag 2. Raws 8 to 11. The authors claims: "When 5 phr of graphite added into nitrile rubber instead of carbon black was added, the storage modulus of the nitrile rubber material was increased accordingly, which make and the effective damping temperature range became widely., which both were beneficial to improved the damping performance.
- The last part of the sentence is unreadable and the message completely obscure;
· Pag 2. Raw 14. The authors are invited to replae "oil palm" with "palm oil";
· Pag 2. Raws 31 to 32. The authors claim: "OMMT also be used in rubber to improve the electrical conductivity of materials [24]".
- The authors are invited to specify in which way the electrical properties are improved. In example, is the conductivity increased or decreased?. In my experience with OMMT, the mentioned properties is decreased. Is this the case the authors are referring to?
· Pag 3. The authors claim: "..., banding the roll for 2 minutes."
- The authors are invited to modify the sentence. Banding???;
· Pag 3. The authors claim: "Next, set the distance between the two mills to 1mm, made triangle bag and roll 5 times respectively"
- The authors are invited to modify the sentence. It is unreadable;
· Pag 5. The authors claim: "It can be seen that the OMMT content has an adversely effects on the physical and mechanical properties".
- Replave with: "It can be seen that the OMMT content adversely effects the physical and mechanical properties"
· Pag 5. Despite the authors deal with the elastic modulus variation with OMMT content, in my opinion the related values have to be included in table 2 (this is mandatory).
· Pag 14. SEM Images. The authors claim: "The SEM image showed that on the surface morphology of the cross-section of OMMT/CB/NR composite was devoid of OMMT particles as shown in Figure 8(a).
- The authors are invited to improve the English language, as their message is not completely understood;
· Pag 14. SEM Images. The authors claim: "Compared with the surface morphology of OMMT/CB/NR composite sample (0 phr), OMMT particles were observed on the surface of the OMMT/CB/NR composite sample (2 phr OMMT) and their distribution was relatively even, as shown in Figure 8(b)".
- The authors are invited to replace even with "uniform" or something similar;
